# The Effect of the Addition of Ozonated and Non-Ozonated Fruits of the Saskatoon Berry (*Amelanchier alnifolia* Nutt.) on the Quality and Pro-Healthy Profile of Craft Wheat Beers

**DOI:** 10.3390/molecules27144544

**Published:** 2022-07-16

**Authors:** Józef Gorzelany, Michał Patyna, Stanisław Pluta, Ireneusz Kapusta, Maciej Balawejder, Justyna Belcar

**Affiliations:** 1Department of Food and Agriculture Production Engineering, University of Rzeszow, 4 Zelwerowicza Street, 35-601 Rzeszów, Poland; gorzelan@ur.edu.pl (J.G.); mpatyna95@gmail.com (M.P.); 2Department of Horticultural Crop Breeding, the National Institute of Horticultural Research, Konstytucji 3 Maja 1/3 Street, 96-100 Skierniewice, Poland; stanislaw.pluta@inhort.pl; 3Department of Food Technology and Human Nutrition, University of Rzeszow, 4 Zelwerowicza Street, 35-601 Rzeszów, Poland; ikapusta@ur.edu.pl; 4Department of Food Chemistry and Toxicology, University of Rzeszow, Ćwiklińskiej 1A Street, 35-601 Rzeszów, Poland; mbalawejder@ur.edu.pl

**Keywords:** Saskatoon berry fruit, ozonation, wheat beers, quality of fruity wheat beers

## Abstract

Research into the suitability of domestic raw materials, including, for example, new wheat cultivars and fruit additives for the production of flavoured beers, is increasingly being undertaken by minibreweries and craft breweries. The fruits of the Saskatoon berry are an important source of bioactive compounds, mainly polyphenols, but also macro- and microelements. The fruits of two Canadian cultivars of this species, ‘Honeywood’ and ‘Thiessen’, were used in this study. Physicochemical analysis showed that wheat beers with the addition of non-ozonated fruit were characterised by a higher ethanol content by 7.73% on average. On the other hand, enrichment of the beer product with fruit pulp obtained from the cv. ‘Thiessen’ had a positive effect on the degree of real attenuation and the polyphenol profile. Sensory evaluation of the beer product showed that wheat beers with the addition of ‘Honeywood’ fruit were characterised by the most balanced taste and aroma. On the basis of the conducted research, it can be concluded that fruits of both cvs. ‘Honeywood’ and ‘Thiessen’ can be used in the production of wheat beers, but the fermentation process has to be modified in order to obtain a higher yield of the fruit beer product.

## 1. Introduction

Wheat beers are beverages for which several raw materials are used, such as: unmalted wheat (e.g., Witbier-style beers), wheat malt (generally 40% to 60% of the raw material charge), barley malt, hops, water and yeast [1,2]. A higher degree of turbidity and a more persistent yet delicate beer head and low bitterness sensation compared to barley beers are characteristics of wheat beers [3,4,5,6]. A typical wheat beer is a beverage fermented using a strain of yeast which is most commonly *Saccharomyces cerevisae*. The colour of wheat beer is light golden and often opaque. Wheat beers are characterised by their original palatability due to the wide range of compounds produced during the fermentation process (including phenols, aldehydes, esters and their derivatives) giving the sensation of vanilla, cloves, banana or fresh fruit, among others [1,3,4,5]. Wheat beers are also characterised by a high content of antioxidant compounds, including polyphenols [7].

Fruit beers have become a summer trend among beverages in recent years, mainly in the form of so-called radlers, that is, beer drinks characterised by the addition of fruit juice or fruit flavour [8,9]. The enrichment of beers with fruit increases the content of bioactive compounds and antioxidant activity of beer beverages, and also determines their sensory qualities, e.g., the taste and aroma of beers [10,11]. The most common beers on the world markets are cherry, raspberry, banana, strawberry or beers with the addition of exotic fruits. Due to the nutritional and health-promoting qualities of the Saskatoon berry (*Amelanchier alnifolia* Nutt.), it can be a valuable addition to the production of fruit beers.

The Saskatoon berry is a shrub belonging to the rose family found in Europe, North America, Africa and the eastern part of Asia [12,13]. It is most widely cultivated in Canada and recently also on a small scale in Finland, the Czech Republic, Lithuania, Latvia and Poland [13,14]. The fruits of the Saskatoon berry are a source of many health-promoting nutrients and can be used as functional food ingredients. They are particularly rich in soluble and insoluble fibre, vitamins such as tocopherol, riboflavin, ascorbic acid, pyridoxine, thiamine and riboflavin, minerals, i.e., manganese, magnesium, iron, calcium and potassium, sugars including sucrose, glucose, fructose and sorbitol, organic acids, protein and pectin. The caloric value of the fruit is averaged at 85 kcal/100 g [12]. The main groups of polyphenols found in the fruit of the Saskatoon berry include flavanols, anthocyanins, flavonols and phenolic acids [15]. The fruit peel of the Saskatoon berry is rich in anthocyanins, including cyanidin derivatives, flavonols and quercetin derivatives. The skin and pulp of the Saskatoon berry are rich in phenolic acids, including chlorogenic acid and neochlorogenic acid. Compared to other fruits, the Saskatoon berry has a 20% higher antioxidant content than cranberries [16], while a 40% higher content compared to aronia (*Aronia melanocaroa* L.) [17]. Other health-promoting properties of the fruit are related to the content of carotenoids and triterpenoids, which show anti-inflammatory effects [18,19]. Consumption of the Saskatoon berry has positive effects on vision, the cardiovascular system and contributes to a lower blood pressure [8].

A factor that can positively influence the production process and the fruit beer’s quality is ozonation fruit and then their being added to the fermentation wort. Ozone is a chemical with strong oxidising properties that causes the disinfection of plant raw material subjected to the process, thus extending its technological shelf life [20]. The antimicrobial action of ozone influences the reduction of microbiological infections during the fermentation of beers with ozone-treated fruit, which has a significant impact both on the fermentation process itself and on the quality of the finished product (taste and aroma). Ozonation of fruits can be performed both before harvest (reducing the occurrence of diseases; e.g., grey mould—*Botrytis cinerea*) and at particular stages of raw material processing (improving processing properties [8]). The use of ozone treatment has a positive effect in the reduction of water losses during fruit storage, increasing antioxidant activity or reducing the release of ethylene by treated fruit. Ozone can be used in two forms: aqueous or gaseous, but studies on fruit ozonation have shown a better efficiency of the process with the latter form [21,22,23]. 

The purpose of this study was to determine the physicochemical properties, sensory properties and antioxidant activity of wheat beers with the addition of ozonated and non-ozonated Saskatoon berry fruits, and to determine the possibility of the practical application of the research results to expand the range of fruit beers and to use these fruits in a new food industry.

## 2. Results and Discussion

### 2.1. Physicochemical Characteristics of Fruit Wheat Beers

Fruit beers should be characterised by the colour of the finished product coming from the added fruit and show good sensory and health-promoting qualities. The results on the evaluation of the physical and chemical parameters of wheat beers are presented in Table 1.

Wheat beers enriched with Saskatoon berry fruit pulp were characterised by an apparent extract of 4.09–4.71% m/m and a real extract of 4.82–5.41% m/m and were, on average, 24.94% and 6.01% higher than wheat beers without these fruit (CB; Table 1). Compared to barley beers enriched with Saskatoon berry fruit [8], wheat beers were characterised by a higher apparent extract by 33.3% on average and a real extract by 7.36% on average, which affected the attenuation of beer and a lower ethanol content. 

As reported by Mascia et al. [24], beer attenuation significantly influenced the ethanol content, which was a determining factor in the beer beverages (content of alcohol) and in the taste and aroma profile of the finished product. In our study, the highest apparent attenuation among beers enriched with Saskatoon berry fruit was characterised by HB0 beer, while the real attenuation was characterised by TB1 beer. All fruit wheat beers were characterised by lower values of the assessed parameters (14.26% and by 9.62%, respectively) in relation to the control wheat beer—CB (Table 1). The lower attenuation of the fruit beers affected the ethanol content, which ranged from 4.04% *v*/*v* to 4.48% *v*/*v*. Beers with the addition of non-ozonated fruits of the Saskatoon berry were characterised by a higher ethanol content of 7.73% on average in comparison with beers enriched with ozonated fruits of this species (Table 1). Fruity barley beers (enriched with the pulp of the Saskatoon berry) studied by Gorzelany et al. [8] were characterised by a higher ethanol content of 5.03% *v*/*v* on average. In cherry and blueberry fruit-enriched beers, Yang et al. [11] investigated apple beer and cranberry beer and reported ethanol contents of 3.5% *v*/*v* and 3.6% *v*/*v*, respectively. The ethanol content in raspberry fruit beer was between 2.8–3.5% *v*/*v* [9]. In the study by Baigts-Allende et al. [10], cherry beers were characterised by an alcohol content in the range of 3.2–8.0% *v*/*v*, with raspberries being 2.5–5.7% *v*/*v* and blackcurrants being 7.1% *v*/*v*. In contrast, Nedyalkov et al. [25] obtained an ethanol content of 5.13% *v*/*v* in the barley beer with bilberry, whereas beers produced with the addition of mango juice and pulp were found with a calorific value of 34.13–36.73 kcal and alcohol contents of 4.13–4.27% *v*/*v* [26]. The fruit wheat beers’ characterised caloric content of the finished product was at the level of 48.88–52.34 kcal/100 mL (Table 1). The caloric content of the barley beers enriched with the pulp of the ozonated and non-ozonated fruits of the Saskatoon berry was lower and averaged 44.83 kcal/100 mL [8].

The addition of pulp from Saskatoon berry fruits to wheat beers significantly affected the colour of the finished product and the process of the ozonating fruits reduced the colour intensity of the colour of the fruit beers by 8.01% on average, compared to wheat beers enriched with non-ozonated fruits (Table 1; Figure 1 and Figure 2). The most intensive colour among the fruit wheat beers was characterised by beer enriched with the non-ozonated fruits of the Saskatoon berry cultivar ‘Thiessen’ (TB1). In the study by Gorzelany et al. [8], the average colour intensity of the barley beers enriched with fruits of the Saskatoon berry (ozonated and non-ozonated fruits) was 22.18 EBC units. Beers with added blackcurrants were characterised by a colour of 14.97 EBC units [10]. In the study by Patraşcu et al. [9], beers with raspberry fruit were marked by a colour of 21.16 EBC units.

Compared to the control wheat beer (CB), the fruit beers were characterised by a slightly higher acidity of the finished product, especially those enriched with non-ozonated fruits of the Saskatoon berry, cultivar ‘Thiessen’, designated as TB1 (Table 1). In addition, the fruits of the Saskatoon berry cultivar ‘Thiessen’ not subjected to the ozonation process were characterised by the highest acidity, both for fruits subjected to the ozonation process and fruits of the cultivar ‘Honeywood’ (ozonated and non-ozonated fruit). The pH value of all fruit beers and the control beer (CB) was at a similar level, from 4.40 to 4.54 (Table 1). In the study by Gorzelany et al. [8], the pH of barley beers (with the addition of Saskatoon berry fruits) averaged 4.53, and the acidity of the finished beer product was 2.2–2.3. In the study by Patraşcu et al. [9], beers with raspberry fruit were characterised by acidity and pH, respectively: 2.84–3.50 and 4.24. Nardini and Garaguso [27], when analysing fruit beers, found that the pH was in the range of 3.56–4.86. Adadi et al. [28] reported the pH value and acidity of beer enriched with sea-buckthorn berries amounting to 3.9 and 2.2, respectively. It is worth noting that the lower the pH of the finished beer product, the lower the risk of infection and development of undesirable microflora [26]. Similarly, fruits subjected to the ozonation process, which destroys or inactivates the microflora present in the fruit peel, may constitute a safer batch added on the seventh day of beer wort fermentation from the point of view of reducing microbiological risk.

In our studies, the addition of Saskatoon berry fruit reduced the perception of bitterness in wheat beers. The finished product enriched with ozonated fruits was characterised by a lower bitterness content of 4.35% in comparison with wheat beers with the addition of non-ozonated fruits of the Saskatoon berry (Table 1). The main factor that affects the bitterness sensation in beers is the used cultivar, its dose and the content of chemical compounds, including α-acids. The boiling time with hops is also an important factor, on which the rate of the protein–polyphenol reaction also depends [3,29]. The reduction in the bitterness sensation in fruity wheat beers is also related to the addition of pulp from the fruit of the Saskatoon berry, which is characterised by a relatively high sugar content; on average it was 14.78 g/100 g d.m., depending on the Saskatoon berry cultivar [18]. In our study, the carbon dioxide content of wheat beers, with or without the addition of Saskatoon berry fruit, ranged from 0.42% to 0.47% (Table 1). Gorzelany et al. [8] obtained a similar carbon dioxide content from barley beers with an addition of these fruits. Patraşcu et al. [9] reported contents of carbon dioxide in lemon beer samples in the range of 0.48–0.55%, in grapefruit beer amounting to 0.52% and in cranberry beer amounting to 0.55%.

### 2.2. Content of Bioactive Compounds in Fruit Beers

Beers are beverages, mostly alcoholic, but at the same time, they contain in their composition compounds of an antioxidant nature, the main representatives of which are polyphenols, but also vitamins, melanoids or bitter acids [30,31]. The antioxidant activity (determined by three methods: DPPH^.^, FRAP and ABTS^+^) of wheat beers enriched with pulp from non-ozonated and ozonated fruits of the Saskatoon berry was presented in Table 2.

The higher antioxidant activity of wheat beers determined by the DPPH and ABTS methods was found in fruit beers, compared to beer without added fruit (control CB). The final products of the fermentation process enriched with non-ozonated Saskatoon berry showed on average 5.78% higher antioxidant activity determined by the DPPH method and 9.55% higher activity determined by the ABTS method compared to beers enriched with pulp from ozonated fruit pulp (Table 2.). Wheat beer without added fruit (CB) showed the highest reducing capacity of the beers (FRAP method). Similarly, as in the case of the antioxidant activity determined by the DPPH and ABTS methods of the analysed beers, finished products with added fruit pulp without ozonation were also characterised by higher activity, on average by 17.22% (Table 2). The antioxidant activity of the used cultivars of Saskatoon berry fruit (determined by the DPPH method) showed that the fruit subjected to the ozonation process was characterised by a slightly higher antioxidant activity in relation to the non-ozonated fruit. Barley beers enriched with Saskatoon berry fruit pulp had slightly lower antioxidant activity, determined by the DPPH and ABTS methods, while higher activity was determined by the FRAP method in relation to wheat fruit beers [8]. At the same time, barley beers showed a positive effect of the addition of ozonated fruits on antioxidant activity, in contrast to wheat beers, whose antioxidant activity was higher when pulp from non-ozonated fruits was added [8]. Deng et al. [32] enhanced beer with omija fruit added during the fermentation process and reported antioxidant activity, measured by a DPPH assay, amounting to 1.68 mM TE/L, and reducing capacity, assessed with FRAP, at a level of 2.4 mM Fe^2+^/L. Portuguese commercial fruit beers with lemon flavour were reported to have an antioxidant capacity in the range of 0.035–0.037 mM TE/L, according to the DPPH assay, and at a level of 0.008 mM TE/L, according to the ABTS assay [33]. The Saskatoon berry fruit, as an addition to the analysed beers, showed very high antioxidant activity, such as for the cultivar ‘Honeywood’—21 mM/100 g d.m. (by the FRAP method) and 31.06 mM/100 g d.m. (by ABTS^+^ method), and for the cultivar ‘Thiessen’—32.32 mM/100 g d.m. (determined by the ABTS^+^ method [15,18].

Polyphenolic compounds present in beers are mainly derived from the malt (70–80%) and the hops used [34]. The degree of fineness of the malt, as well as the conditions of the mashing and boiling process with hops, significantly affect the total polyphenol content [29]. Polyphenolic compounds are diverse substances with different biologically active effects, including antioxidant and antiradical activity [35]. We confirmed that the addition of Saskatoon berry fruits to wheat beer increased the content of the total polyphenols, on average by 37.37% compared to beer without the addition of fruits (CB). The differences in the content of polyphenolic compounds in fruit beers (addition of non-ozonated and ozonated fruit) were statistically significant (Table 3). The degree of the transfer of phenolic compounds contained in the fruit to wheat beer depends on the degree of grinding of the fruit. The use of fruit pulp as an input to the fermenting wort increases the contact with the solution, which leaches and transfers the chemical compounds found in the fruit through the disrupted cell wall, thus enriching the finished beer product [26]. According to Gorzelany et al. [8], fruity barley beers with the addition of the Saskatoon berry had an average total polyphenol content of 381 mg GAE/L for beers enriched with non-ozonated fruit and 388 mg GAE/L for beers with the addition of ozonated fruit. The data from the literature showed that the total polyphenol content of the beers enriched with Cornelian cherry was 350 mg GAE/L [36] and with goji berries was 415 mg GAE/L [37]. The addition of persimmon juice led to a decrease in the total polyphenol content in the beer samples from 433.32 mg GAE/L (25% juice addition) to 290.34 mg GAE/L (75% juice addition; [34]). Portuguese commercial fruit beers with lemon flavour were found with total polyphenol contents in the range of 240–304 mg GAE/L [33]. 

The identification of polyphenolic compounds in fruity wheat beers was based on the analysis of characteristic spectral data: the mass-to-charge ratio m/z and the maximum of radiation. In our study, a total of 11 polyphenolic compounds were identified, whose spectral properties are presented in Table 3. In control wheat beer (CB), three polyphenolic compounds belonging to the flavonol group (compounds 1–3) were identified. Their representatives were kaempferol derivatives, of which the highest concentration (1.35 mg/L) was determined for K-3-*O*-glucoside-7-*O*-glucoside (Table 3). In wheat beers enriched with fruits of the Saskatoon berry, eight compounds belonging to the group of hydroxycinnamic acid derivatives (compounds 4–7; 11) and to the group of flavonols (compounds 8–10) were identified. The content of the polyphenolic compounds in wheat beers enriched with ozonated Saskatoon berry fruits was, on average, 7.74 mg/L, while with the addition of non-ozonated fruits, it was slightly higher and amounted, on average, to 8.87 mg/L (Table 3). The results of the present study confirmed the effect of ozonation on the reduction of the content of polyphenolic compounds in beers with the addition of fruits of the Saskatoon berry, which was previously obtained by Gorzelany et al. [8]. The lower concentration of polyphenolic compounds in ozonated fruit-enriched beers is most likely related to the interaction between the ozone remaining on the fruit and the products of the fermentation process, but this hypothesis is not fully understood. However, there are results in the international literature that confirm the positive effect of ozone on the polyphenolic profile of different fruits [20,38,39,40,41].

Among the phenolic acids present in wheat beers, the content of the chlorogenic acid content was on average 63.09% higher in beers enriched with non-ozonated fruits of the Saskatoon berry. The highest concentration of this acid (2.17 mg/L) was determined for beer with the addition of non-ozonated fruits of the ‘Honeywood’ cultivar, designated as HB1 (Table 3). Chlorogenic acid was present in ozonated fruit wheat beers, in contrast to ozonated fruit barley beers in which its presence was not detected. In contrast, the main representative of the polyphenolic compounds in barley beers with added fruit was caffeic acid, whose concentration averaged 4.01 mg/L [8]. In the wheat beers with the addition of Saskatoon berry fruit, the caffeic acid was between 0.57 and 0.96 mg/L (Table 3). From the health point of view, caffeic acid is responsible for blocking some substances with carcinogenic effect, e.g., nitrosamines, and it also affects the oxidation process of lipoproteins and LDL cholesterol fraction [42]. The caffeic acid in beers is generally in the range of 0.00–23.50 mg/L [43]. In barley beers with bilberry (fruit addition in the amount of 167 g/L), the content of caffeic acid was 13.01 mg/L, chlorogenic acid was 90.19 mg/L and neochlorogenic acid was 52.24 mg/L [25]. Wheat beers enriched with fruits of the cv. ‘Thiessen’ of the Saskatoon berry were further characterised by a high concentration of sinapic acid derivative; on average, it was 50.0% more than in fruits of the cv. ‘Honeywood’ (Table 3). Kaempferol glycosides have also been identified in fruit wheat beers, which have strong antioxidant, anticancer and supportive properties in cardiovascular diseases. In addition, they can be supportive substances in autoimmune diseases and for transplant patients [44]. In barley beers, kaempferol compounds are most commonly found at 0.10—1.64 mg/L [43]. Flavonoid glycosides (including kaempferol derivatives), as well as chlorogenic acid, caffeic acid or sinapic acid contained in beers impart astringency and acidity sensations in the mouth, as well as, although to a much lower extent, also a bitterness sensation which affects the sensory experience of the finished beer product [44].

### 2.3. Sensory Analysis of Fruit Wheat Beers

The sensory characteristics of a finished fruit beer product have a significant impact on its attractiveness and acceptance by consumers. The taste and aroma qualities of fruity wheat beers can influence consumers’ preference to purchase a particular beer, or this purchase will only be a one-off. The results of the sensory evaluation of fruit wheat beers carried out by a 13-member panel are presented in Table 4 and Figure 3 and Figure 4.

Fruity wheat beers were characterised by a similar smell sensation of the finished product (4.00–4.23 points, on a 5-point rating scale), but with a statistically different taste. The highest taste sensory ratings were obtained for wheat beer enriched with non-ozonated fruit of the cv. ‘Honeywood’ (Table 4, Figure 3) and with ozonated fruit of the cv. ‘Thiessen’ (Table 4., Figure 4). The taste and smell of beer are influenced not only by the raw materials used, but also by the products of the fermentation process (such as aldehydes, phenols, or esters) affecting the taste profile of a given beer. Among the quality attributes of fruity wheat beers, the stability of the beer head was assessed the lowest, especially for beers enriched with the cv. ‘Honeywood’ fruit irrespective of the applied ozonation process (Table 4). Sensory evaluation confirmed the results of the physicochemical analysis regarding the lower bitterness sensation in fruity wheat beers compared to the control beer (CB).

The sensory profile of the wheat fruit beers varied. Sensory evaluation showed that the most balanced flavour profile was characterised by wheat beers enriched with unfermented the cv. ‘Honeywood’ fruit irrespective of the cultivar used (HB1 and TB1; Figure 3 and Figure 4). The significant quality attributes of the fruited wheat beers with the cv. ‘Honeywood’ fruit were cereal and malty, fruity, sweet and sour tastes and aroma (Figure 3.). On the other hand, wheat beers enriched with fruit from the cv. ‘Thiessen’ additionally had a bitter aftertaste, especially for beers with ozonated fruit added (Figure 4). The aftertaste of the sour, astringent or bitter fruit of wheat beers is related to the varying content of the polyphenols responsible, including caffeic acid and chlorogenic acid [29]. In a study on the possibility of enriching barley beers with the Saskatoon berry, Gorzelany et al. [8] obtained similar results for the flavour and aroma profile and a bitter aftertaste was also clearly perceived in the finished product, especially in the finished product enriched with non-ozonated fruit. Chemical compounds important for beer flavour are formed by interactions between carbonyl compounds, esters, sulphur compounds, alcohols, phenolic compounds or organic acids [45]. Beers characterised by fruity notes with a sweet aftertaste and pleasant aroma are more preferred and desired by consumers compared to traditional types of beers [28,46].

## 3. Materials and Methods

### 3.1. Material

Common wheat (*Triticum aestivum* L.), the winter variety ‘Elixer’, was used to produce wheat beers. The grain came from a field experiment collected in the year 2021, in Przeworsk (50°03′31″ N 22°29′37″ E), Podkarpackie Province (south-east Poland). After full maturity, the grain was harvested and a 5-day wheat malt was prepared (the malting process methodology is described by Belcar et al. [47]). The wheat malt had the following characteristics: extract potential—85.7% d.m. (d.m.—dry matter), total protein content—11.6% d.m., content of soluble protein—4.67% d.m., diastatic power—324 WK, and degree of final attenuation—82.14%.

Commercial barley malt from the Viking Malt malting plant in Strzegom (Poland) was also used to brew the beers. The barley malt had the following characteristics: extract potential—80.0% d.m., total protein content—11.4% d.m., content of soluble protein—3.75% d.m., diastatic power—324 WK, and degree of final attenuation—82.1%. The wheat and barley malts were refined in a Cemotec disc mill (FOSS). The brewing stock consisted of 40% wheat malt and 60% commercial barley malt. 

Fruits of two Canadian cultivars of Saskatoon berry ‘Honeywood’ and ‘Thiessen’ were used to enrich the wheat beers. Ripe fruits weighing 2 kg each were harvested by hand from 6-year-old bushes grown in an implementation experiment in a field at the Experimental Orchard in Dąbrowice (51.9163° N/20.1009° E) of the National Institute of Horticultural Research (InHort) in Skierniewice, central Poland, at the beginning of July 2021. In the laboratory of the Department of Agriculture and Food Production Engineering of the University of Rzeszów, the Saskatoon berry fruits were divided into two samples of 1 kg each (one part was left without ozonation and the other part was ozonated). Until beers were produced, ozonated and non-ozonated fruits of Saskatoon berry were directly frozen and stored in a freezer (temp. −18 °C). The non-ozonated Saskatoon fruit cultivar ‘Honeywood’ had the following chemical parameters: total polyphenol content—3.67 g GAE^.^1000 g^−1^ d.m., antioxidant activity (DPPH test)—17.38 mM TE^.^100g^−1^ d.m., and total acidity—0.503 g^.^100 g^−1^, whereas cultivar ‘Thiessen’ had the following chemical parameters: total polyphenol content—5.16 g GAE^.^1000 g^−1^ d.m., antioxidant activity (DPPH test)—20.97 mM TE^.^100g^−1^ d.m., and total acidity—0.963 g^.^100 g^−1^.

### 3.2. Ozonation Process

The Saskatoon berry fruits of both cultivars were placed on a metal grid inside a plastic container with dimensions L × W × H—0.6 × 0.4 × 0.4 m—and ozonated for 22 min—ozone concentration 10 ppm, flow time 4 m^3.^h^−1^, temperature 20 °C. The TS 30 ozone generator (Ozone Solution, Hull, MA, USA) with a 106 M UV Ozone Solution detector (Ozone Solution, Hull, MA, USA) was used to generate ozone. The ozone-treated Saskatoon fruit cultivar ‘Honeywood’ had the following chemical parameters: total polyphenol content—3.81 g GAE^.^1000 g^−1^ d.m., antioxidant activity (DPPH test)—17.40 mM TE^.^100g^−1^ d.m., and total acidity—0.352 g 100 g^−1^, whereas cultivar ‘Thiessen’ had the following chemical parameters: total polyphenol content—3.23 g GAE^.^1000 g^−1^ d.m., antioxidant activity (DPPH test)—21.12 mM TE^.^100g^−1^ d.m., and total acidity—0.691 g^.^100 g^−1^.

### 3.3. Beer Production

The production process was carried out using the infusion method in the laboratory of the Department of Agriculture and Food Production Engineering, University of Rzeszów. Total of 3.0 kg of barley malt and 2.0 kg of wheat malt were grated and placed in a ROYAL RCBM-40N mash kettle (Expondo; Poland; assuming a process efficiency of 80%) and 15.0 L of water (3 L of water for each kilogram of malt). The mashing, boiling process with hops and cooling of the beer wort were carried out according to the methodology described by Gorzelany et al. [8]. 

Each of the five beer worts produced was characterised by an extract of 12.0 °P. The cooled worts were transferred to fermentation containers with a capacity of 30 L each and inoculated with *Saccharomyces cerevisae* Fermentis Safale US-05 yeast (6 × 10^9^/g), which had previously undergone a rehydration process, according to the manufacturer’s instructions (0.58 g d.m./L of wort). The fermentation process was carried out at 21 °C. After 7 days of fermentation, 1 kg of Saskatoon berry fruit was added to the fermenting beer in the form of pulp and left to ferment for another 14 days. After 21 days, the beers were bottled, with a solution of sucrose (0.3%) added to water for refermentation and to obtain an appropriate degree of beer saturation. The resulting beers were kept at 20 °C. Sensory and physicochemical analyses were performed one month after bottling. 

Wheat beers enriched with the cv. ‘Honeywood’ fruit were designated as HB0 (ozonated fruit) and as HB1 (non-ozonated fruit), while wheat beers with the cv. ‘Thiessen’ fruit were designated as TB0 (ozonated fruit) and as TB1 (non-ozonated fruit). The wheat beer without added fruit as a control was designated CB. A total of 5 wheat beers were produced.

### 3.4. Analysis of Quality Indicators of Beers

Alcohol content [% m/m and % *v*/*v*], apparent extract [% m/m], real extract [% m/m] and original extract [% m/m] of beer, and apparent [%] and real [%] attenuation degree was marked according to method 9.4 EBC [48]. The titratable acidity of fruit wheat beers was determined by subjecting beer samples to titration with 0,1 M NaOH, with end point at pH = 8.2. The energy value of wheat beers was calculated following the formula: [kcal/100 mL] = (7 × A (% *v*/*v*) + (4 × Er (% *v*/*v*) × ρ). pH, colour [EBC units], carbon dioxide content [%] and bitterness content [IBU units] of beer were determined according to the methodology described by Belcar et al. [1]. The analyses were performed in three replications.

### 3.5. Content of Bioactive Compounds in Fruit Beers

The total polyphenol content [mg GAE/L], by using the Folin–Ciocalteu method, and the polyphenol profile [mg/L] in the beers were determined according to the methodology described by Gorzelany et al. [8]. Determination of polyphenolic compounds [mg/L] was carried out using the UPLC equipped with a binary pump, column and sample manager, photodiode array detector (PDA), and tandem quadrupole mass spectrometer (TQD) with electrospray ionisation (ESI) source working in negative mode (Waters, Milford, MA, USA), according to the method of Żurek et al. [49]. Separation was performed using the UPLC BEH C18 column (1.7 µm, 100 mm × 2.1 mm, Waters) at 50 °C, at flow rate of 0.35 mL/min. The injection volume of the samples was 5 µL. The mobile phase consisted of water (solvent A) and 40% acetonitrile in water, *v*/*v* (solvent B). The following TQD parameters were used: capillary voltage of 3500 V, con voltage of 30 V, con gas flow 100 L/h, source temperature 120 °C, desolvation temperature 350 °C and desolvation gas flow rate of 800 L/h. Polyphenolic identification and quantitative analyses were performed on the basis of the mass-to-charge ratio, retention time, specific PDA spectra, fragment ions and comparison of data obtained with commercial standards and literature findings. The analyses were performed in three replications. 

### 3.6. Antioxidant Activity 

The antioxidant activity of fruit beers (by DPPH^.^ [mM TE/L], FRAP [mM Fe^2+^/L] and ABTS^+^ [mM TE/L]) was determined according to the methodology described by Gorzelany et al. [8]. 

#### 3.6.1. DPPH Test

A 0.05 mM/L solution of DPPH (2,2-diphenyl-1-picrylhydrazyl) in ethanol was prepared for this purpose. A 7.8 mL sample of the solution was placed in a test tube with 0.2 mL of diluted (2×) beer and incubated in darkness for 60 min at 37 °C; subsequently, the absorbance at the wavelength λ = 517 nm was examined using a UV-Vis V-5000 spectrophotometer (Shanghai Metash Instruments Co. Ltd., Shanghai, China). The control contained distilled water rather than beer. The results were expressed as trolox equivalent (mM TE/L).

#### 3.6.2. FRAP Test

The materials prepared for this purpose included a 10 mM/L TPTZ (2,4,6-tripyridyl-s-triazine) solution, a 20 mM/L FeCl_3_^.^6H_2_O solution, an acetate buffer with pH = 3.6, as well as a 40 mM/L HCl solution. Subsequently, the FRAP reagent was prepared by mixing 25 mL of the acetate buffer with 2.5 mL of the TPTZ dissolved in HCl and 2.5 mL of FeCl_3_^.^6H_2_O. A 6 mL sample of FRAP solution was placed in a test tube with 0.2 mL of the beer and incubated at a temperature of 37 °C for 10 min; subsequently, the absorbance at the wavelength of λ = 593 nm was examined using a UV-Vis V-5000 spectrophotometer (Shanghai Metash Instruments Co. Ltd., Shanghai, China). The control contained distilled water instead of beer. The results of the FRAP test were expressed as mM Fe^2+^/L.

#### 3.6.3. ABTS Test

A 7 mM/L ABTS (2,2’-azinobis(3-ethylbenzothiazoline-6-sulfonic acid) solution and a 2.45 mM/L potassium persulphate solution were prepared for this purpose. The solutions were combined at 1:0.5 ratio, and stored for 12–16 h in darkness to enable development of ABTS cation. The ABTS^+^ solution was diluted with distilled water to achieve absorbance of 0.700 ± 0.002 (at wavelength λ = 734 nm). A 3 mL portion of the diluted ABTS^+^ solution was placed in the test tube with 0.3 mL of the beer and after 6 min, the absorbance value at the wavelength λ = 734 nm was determined using a UV-Vis V-5000 spectrophotometer (Shanghai Metash Instruments Co. Ltd., Shanghai, China). The results were corrected to account for dilution and expressed as trolox equivalent (mM TE/L).

All the analyses were performed in three replications.

### 3.7. Sensory Analysis

The sensory analysis was performed by an expert team of 13 persons (6 women and 7 men, 25–40 years old) in the sensory evaluation laboratory according to the EBC method 13.13 [50]. The beer samples (of 200 mL each) were served after cooling to 12 °C, coded in random order in 250 mL transparent plastic cups. Oral rinse water was administered between each evaluation. The sensory analysis of the beers was performed using a 5-grade rating scale for individual quality attributes: aroma (5—very intense, distinct, pleasant; 1—undetectable/unpleasant aroma), flavour (5—very tasty; 1—unpalatable); beer foam stability (5—very stable; 1—unstable), bitterness (5—weakly intense; 1—very intense) and saturation (5—high; 1—low or none). The average score obtained described the overall impression of the beer evaluated (5—very good; 1—bad) of the wheat beers analysed. Furthermore, a sensory profile was used to assess the taste and aroma of the beers’ analyses, in which quality characteristics were determined (malty, fruity, sweet, cereal, intense, fullness, fresh, phenolic, bitter and sour) according to the EBC 13.12 method [51]. The sensory profile of fruit beers produced with the addition of non-ozonated and ozonated fruits of the Saskatoon berry was compared with beers without the addition of fruit (control).

### 3.8. Statistical Analysis

The results of the fruit beers were presented as a mean value with standard deviation. Statistical analysis of the results was performed using Statistica 13.3 statistical software (TIBCO Software Inc., Tulsa, OK, USA). Two-factor ANOVA of variance ANOVA was used in the analyses in a complete randomised design with a significance level of α = 0.05 for the individual results of physical and chemical analysis, polyphenol content and antioxidant activity of the fruit beers. Comparisons of mean values were done using the HSD-Tukey test.

## 4. Conclusions

Fruity wheat beers enriched with Saskatoon berry fruit pulp are characterised by a higher colour intensity and lower bitterness sensation, but at the same time insufficient attenuation, affecting the ethanol content. The antioxidant activity of wheat beers and the content of the total polyphenols for both tested cultivars of the Saskatoon berry were at a similar level. However, analysis of the polyphenol profile showed a significantly higher content of polyphenolic compounds in wheat beers enriched with non-ozonated fruits of the cv. ‘Thiessen’. The results of the sensory evaluation show that wheat beers with the addition of the cv. ‘Honeywood’ fruit are characterised by the most balanced taste and aroma. On the basis of the research results obtained, we can conclude that fruits of both cvs. ‘Honeywood’ and ‘Thiessen’ can be used in the production of wheat beers. However, unlike barley beers, the fermentation process has to be modified in order to obtain a higher yield of the fruit beer product, and the ozonation process had a positive effect on improving the quality of fruit wheat beers.

## Figures and Tables

**Figure 1 molecules-27-04544-f001:**
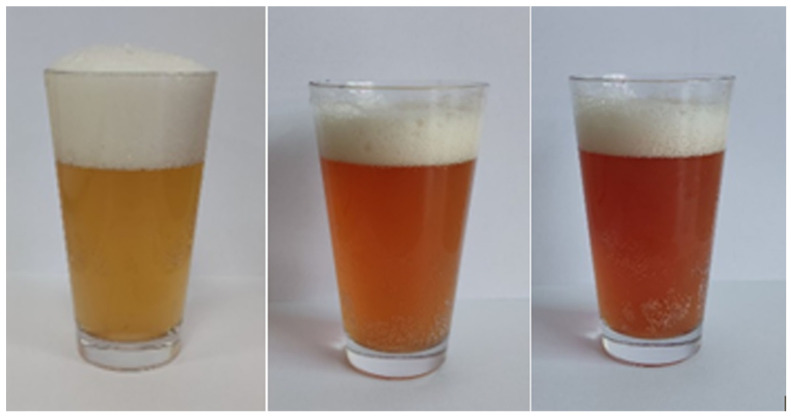
The appearance of the obtained wheat beers with addition of the Saskatoon berry fruits (from left to right)—CB—control beer, HB0—cv. ‘Honeywood’ with ozone-treated fruit and HB1—untreated fruit.

**Figure 2 molecules-27-04544-f002:**
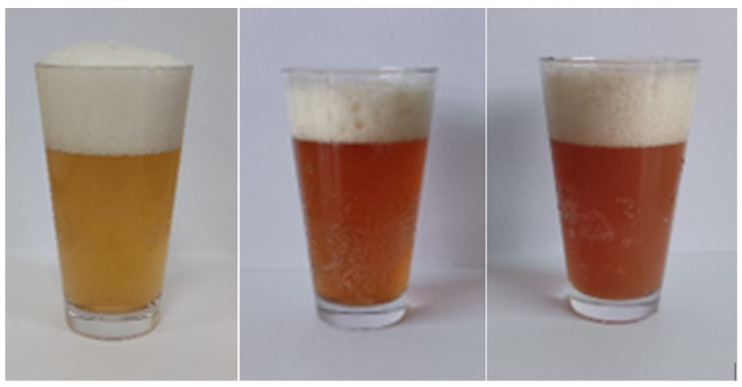
The appearance of the obtained wheat beers with addition of the Saskatoon berry fruits (from left to right)—CB—control beer, TB0—cv. ‘Thiessen’ with ozone-treated fruit and TB1—untreated fruit.

**Figure 3 molecules-27-04544-f003:**
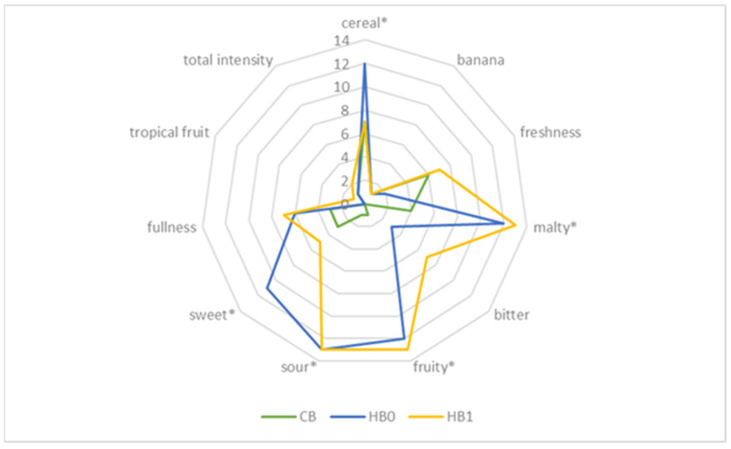
Sensory profile of wheat beers—control (CB) and sample with addition of the cv. ‘Honeywood’ fruit untreated (HB1) and treated with ozone (HB0) (* marks the attributes which were statistically different at *p* ≤ 0.05).

**Figure 4 molecules-27-04544-f004:**
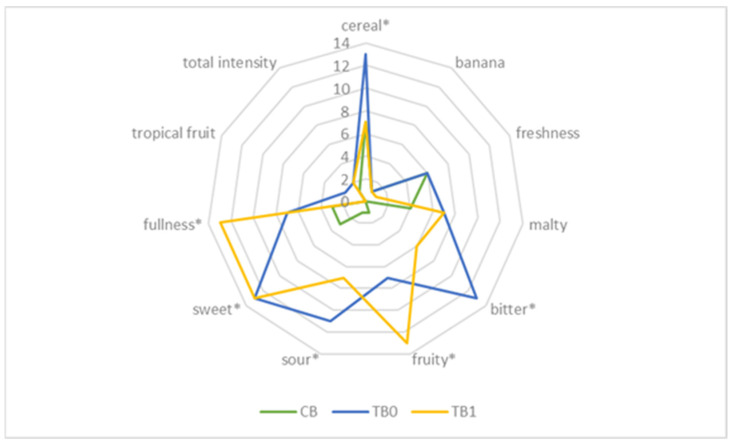
Sensory profile of wheat beers—control (CB) and sample with addition of the cv. ‘Thiessen’ fruit untreated (TB1) and treated with ozone (TB0) (* marks the attributes which were statistically different at *p* ≤ 0.05).

**Table 1 molecules-27-04544-t001:** Physicochemical analysis of the beers produced with an addition of the Saskatoon berry fruit.

Type of Beer	CB	HB0	HB1	TB0	TB1
Apparent extract [%; m/m]	3.31 ^a^ ± 0.09	4.53 ^c^ ± 0.03	4.09 ^b^ ± 0.04	4.71 ^d^ ± 0.01	4.42 ^c^ ± 0.02
Real extract [%; m/m]	4.85 ^a^ ± 0.05	5.35 ^c^ ± 0.05	5.41 ^c^ ± 0.01	5.05 ^b^ ± 0.05	4.82 ^a^ ± 0.04
Original extract [%; m/m]	14.88 ^e^ ± 0.06	13.21 ^b^ ± 0.06	13.70 ^d^ ± 0.10	12.84 ^a^ ± 0.04	13.43 ^c^ ± 0.03
Degree of final apparent attenuation [%]	77.64 ^e^ ± 0.06	65.71 ^b^ ± 0.07	70.14 ^d^ ± 0.06	63.32 ^a^ ± 0.02	67.09 ^c^ ± 0.09
Degree of final real attenuation [%]	67.71 ^d^ ± 0.06	59.50 ^a^ ± 0.50	60.51 ^b^ ± 0.07	60.67 ^b^ ± 0.03	64.11 ^c^ ± 0.03
Content of alcohol [%; m/m]	5.24 ^d^ ± 0.04	4.08 ^a^ ± 0.07	4.32 ^b^ ± 0.02	4.04 ^a^ ± 0.04	4.48 ^c^ ± 0.04
Content of alcohol [%; *v*/*v*]	4.18 ^d^ ± 0.04	3.24 ^a^ ± 0.04	3.44 ^b^ ± 0.04	3.21 ^a^ ± 0.01	3.56 ^c^ ± 0.10
Colour [EBC units]	20.1 ^a^ ± 0.3	23.1 ^b^ ± 0.0	25.2 ^d^ ± 0.2	24.0 ^c^ ± 0.0	26.9 ^e^ ± 0.1
Titratable acidity [0.1M NaOH/100 mL]	3.46 ^a^ ± 0.06	3.55 ^b^ ± 0.05	3.64 ^c^ ± 0.03	3.71 ^c^ ± 0.02	4.22 ^d^ ± 0.03
pH	4.54 ^b^ ± 0.06	4.41 ^a^ ± 0.08	4.42 ^a^ ± 0.02	4.40 ^a^ ± 0.10	4.47 ^b^ ± 0.03
Content of carbon dioxide [%]	15.4 ^d^ ± 0.2	13.9 ^c^ ± 0.4	14.2 ^c^ ± 0.1	12.5 ^a^ ± 0.1	13.4 ^b^ ± 0.0
Bitter substances [IBU]	0.46 ^a^ ± 0.06	0.44 ^a^ ± 0.03	0.46 ^a^ ± 0.02	0.42 ^a^ ± 0.02	0.47 ^a^ ± 0.00
Energy value [kcal/100 mL]	57.22 ^e^ ± 0.02	50.41 ^b^ ± 0.06	52.34 ^d^ ± 0.61	48.88 ^a^ ± 0.13	51.01 ^c^ ± 0.08

Data are expressed as mean values (*n* = 3) ± SD; SD—standard deviation. Mean values within rows with different letters are significantly different (*p* < 0.05). ^a,b,c,d,e^—statistically significant differences for the effect: physicochemical properties of beer × type of beer. CB—control wheat beer; HB—cv. ‘Honeywood’; TB—cv. ‘Thiessen’; 0—wheat beer with treated Saskatoon berry fruit; 1—wheat beer with non-treated Saskatoon berry fruit.

**Table 2 molecules-27-04544-t002:** Antioxidant potential of fruit beers with Saskatoon berry fruit pulp added.

Type of Beer	CB	HB0	HB1	TB0	TB1
DPPH^.^[mM TE/L]	2.27 ^a^ ± 0.07	2.34 ^b^ ± 0.04	2.94 ^d^ ± 0.01	2.71 ^c^ ± 0.07	2.42 ^b^ ± 0.02
FRAP [mM Fe^2+^/L]	2.19 ^d^ ± 0.04	1.46 ^a^ ± 0.06	1.97 ^c^ ± 0.03	1.52 ^a^ ± 0.02	1.63 ^b^ ± 0.03
ABTS^+^ [mM TE/L]	1.81 ^a^ ± 0.05	2.02 ^b^ ± 0.02	2.18 ^c^ ± 0.02	1.96 ^b^ ± 0.04	2.22 ^c^ ± 0.08

Data are expressed as mean values (*n* = 3) ± SD; SD—standard deviation. Mean values within rows with different letters are significantly different (*p* < 0.05). ^a,b,c,d^—statistically significant differences for the effect: antioxidant activity of beer × type of beer. CB—control wheat beer; HB—cv. ‘Honeywood’; TB—cv. ‘Thiessen’; 0—wheat beer with treated Saskatoon berry fruit; 1—wheat beer with non-treated Saskatoon berry fruit; TE—expressed as Trolox equivalent (mM TE/L).

**Table 3 molecules-27-04544-t003:** Contents of polyphenols and polyphenolic profile identified by UPLC-PDA-TQD-MS in wheat beer.

Type of Beer						CB	HB0	HB1	TB0	TB1
Contents of polyphenols [mg GAE/L]						243.90 ^a^ ± 1.85	382.83 ^c^ ± 0.92	413.43 ^d^ ± 0.76	383.65 ^c^ ± 0.43	377.86 ^b^ ± 0.46
No.	Compound [mg/L]	Rt [min]	[M-H]^−^ (*m*/*z*)	Fragment ions (*m*/*z*)	Absorbance maxima (nm)					
1.	K-3-*O*-sophoroside	3.97	609	285	264, 324	0.95 ± 0.02	n.d.	n.d.	n.d.	n.d.
2.	K-3-*O*-rut-7-*O*-glc	4.09	755	593, 285	264, 324	0.92 ± 0.02	n.d.	n.d.	n.d.	n.d.
3.	K-3-*O*-glc-7-*O*-glc	4.20	609	447, 285	264, 324	1.35 ± 0.04	n.d.	n.d.	n.d.	n.d.
4.	Neo-chlorogenic acid	2.88	353	191	299sh, 324	n.d.	0.71 ^a^ ± 0.08	1.21 ^c^ ± 0.12	0.82 ^a^ ± 0.14	1.07 ^b^ ± 0.09
5.	Chlorogenic acid	3.54	353	191	299sh, 324	n.d.	0.83 ^a^ ± 0.08	2.17 ^c^ ± 0.36	0.51 ^a^ ± 0.05	1.46 ^b^ ± 0.01
6.	Sinapic acid glucoside	4.04	385	223	299sh, 326	n.d.	1.63 ^b^ ± 0.04	1.05 ^a^ ± 0.09	2.18 ^c^ ± 0.01	2.23 ^c^ ± 0.26
7.	Caffeic acid	4.13	179	161	299sh, 327	n.d.	0.57 ^a^ ± 0.00	0.96 ^b^ ± 0.01	0.90 ^b^ ± 0.10	0.87 ^b^ ± 0.09
8.	K-3-*O*-glc-pent	4.40	579	285	264, 350	n.d.	0.76 ^b^ ± 0.02	0.66 ^a^ ± 0.08	0.73 ^b^ ± 0.07	0.80 ^b^ ± 0.02
9.	K-3-*O*-rut	4.51	593	285	264, 350	n.d.	0.79 ^a^ ± 0.02	0.78 ^a^ ± 0.08	0.92 ^b^ ± 0.02	0.81 ^a^ ± 0.00
10.	K-3-*O*-rha-7-*O*-pent	4.63	563	431, 285	264, 344	n.d.	1.00 ^b^ ± 0.03	0.97 ^ab^ ± 0.02	1.03 ^bc^ ± 0.00	0.94 ^a^ ± 0.02
11.	Ferulic acid derivative	5.42	610	193	299sh, 327	n.d.	1.09 ^c^ ± 0.05	0.79 ^a^ ± 0.00	1.02 ^b^ ± 0.02	1.00 ^b^ ± 0.03
Total						3.22 ^a^ ± 0.08	7.37 ^b^ ± 0.05	8.57 ^c^ ± 0.69	8.10 ^bc^ ± 0.34	9.16 ^cd^ ± 0.40

Data are expressed as mean values (*n* = 3) ± SD; SD—standard deviation. Mean values within rows with different letters are significantly different (*p* < 0.05). ^a,b,c,d^—statistically significant differences for the effect: contents of polyphenols and polyphenolic profile of beer × type of beer. CB—control wheat beer; HB—cv. ‘Honeywood’; TB—cv. ‘Thiessen’; 0—wheat beer with treated Saskatoon berry fruit; 1—wheat beer with non-treated Saskatoon berry fruit. K—kaempferol; glc—glucoside; rut—rutinoside; pent—pentoside; rha—rhamnoside; GAE—equivalent of gallic acid (mg GAE/L).

**Table 4 molecules-27-04544-t004:** Sensory analysis of fruit wheat beer.

Type of Beer	CB	HB0	HB1	TB0	TB1
Aroma	4.20 ^a^ ± 0.38	4.23 ^a^ ± 0.83	4.15 ^a^ ± 0.99	4.23 ^a^ ± 0.73	4.00 ^a^ ± 1.00
Taste	3.79 ^a^ ± 0.27	4.08 ^ab^ ± 1.34	4.69 ^b^ ± 0.48	4.54 ^b^ ± 0.66	4.08 ^ab^ ± 0.76
Foam stability	3.51 ^b^ ± 0.17	2.46 ^a^ ± 0.97	2.69 ^a^ ± 0.48	3.00 ^ab^ ± 0.58	3.08 ^ab^ ± 0.86
Bitterness	4.06 ^a^ ± 0.11	3.62 ^a^ ± 0.87	3.92 ^a^ ± 0.86	3.77 ^a^ ± 0.93	3.46 ^a^ ± 0.52
Saturation	3.71 ^a^ ± 0.32	4.31 ^ab^ ± 0.63	4.84 ^c^ ± 0.38	4.15 ^ab^ ± 0.80	4.38 ^bc^ ± 0.65
Overall impression	3.91 ^a^ ± 0.47	3.82 ^a^ ± 0.75	4.18 ^a^ ± 0.48	4.11 ^a^ ± 0.49	3.83 ^a^ ± 0.57

Data are expressed as mean values (*n* = 3) ± SD; SD—standard deviation. Mean values within rows with different letters are significantly different (*p* < 0.05). ). ^a,b,c^—statistically significant differences for the effect: sensory analysis of beer × type of beer. CB—control wheat beer; HB—cv. ‘Honeywood’; TB—cv. ‘Thiessen’; 0—wheat beer with treated Saskatoon berry fruit; 1—wheat beer with non-treated Saskatoon fruit.

## Data Availability

Not applicable.

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
