# Peer review of "The Effect of the Addition of Ozonated and Non-Ozonated Fruits of the Saskatoon Berry (Amelanchier alnifolia Nutt.) on the Quality and Pro-Healthy Profile of Craft Wheat Beers"

_molecules, 2022, doi:10.3390/molecules27144544_

Round 1

Reviewer 1 Report

The study of Gorzelany et al. is interesting. The authors presented in a pleasant manner the results of their research on the possibility to obtain wheat beers with the addition of ozonated and non-ozonated Saskatoon berry fruits.

The Latin name of the Saskatoon berry should be indicated only once, when first used in the manuscript body.

Lines 76-77: The statement needs revision. Please check the “…the beer’s quality of the obtained with …”

Line 81: I suggest avoiding the use of the term “ozonized”. More appropriate alternatives are ozone-treated or ozonated.

Line 113: It is not clear what the authors meant by “determining factor in beer beverages”

Lines 129-131: Rephrasing is needed. The main idea is difficult to follow.

Line 144: Avoid the repetition (“batches of beer batches”).

The authors should spend more effort in trying to explain the differences in the equivalent beer samples prepared with native and ozone treated Saskatoon berry.

The study would have been benefited by the detailed characterization of the Saskatoon berry prior and after ozone treatment.

Line 397: The correct terms are: apparent and real attenuation degree.

Author Response

Authors are grateful for the contribution of the Reviewer.

The Latin name of the Saskatoon berry should be indicated only once, when first used in the manuscript body.

Lines 76-77: The statement needs revision. Please check the “…the beer’s quality of the obtained with …”

Line 81: I suggest avoiding the use of the term “ozonized”. More appropriate alternatives are ozone-treated or ozonated.

Line 113: It is not clear what the authors meant by “determining factor in beer beverages”

Lines 129-131: Rephrasing is needed. The main idea is difficult to follow.

Line 144: Avoid the repetition (“batches of beer batches”).

Line 397: The correct terms are: apparent and real attenuation degree.

Answer:

All comments were corrected in manuscript

The study would have been benefited by the detailed characterization of the Saskatoon berry prior and after ozone treatment.

Answer:

Chemical properties of Saskatoon fruit before and after the ozonation process were added to the material (lines 414-419 and lines 426-431)

The authors should spend more effort in trying to explain the differences in the equivalent beer samples prepared with native and ozone treated Saskatoon berry.

Answer:

More information has been added to the manuscript regarding the differences in wheat beers enriched with ozonated and non-ozonated Saskatoon fruits (f.ex. lines 242-245; 186-189).

Reviewer 2 Report

Comments to the Editors/Authors:

This is a study on "The effect of the addition of ozonated and non-ozonated fruits of the Saskatoon berry (Amelanchier alnifolia Nutt.) on the quality and pro-healthy profile of craft wheat beers" that is similar to their previous study on barley beers and from this side I think has low originality.

Nevertheless, this work might be improved by taking into account the following suggestions:

ABSTRACT: Please reduce the number of words up to 200 according to the journal’s instructions.

Please omit the “.” After “7th”

INTRODUCTION:

Line 65: Please use the same unit methodology and write “85 Kcal / 100 g”

Line 70: “other health-promoting properties” and writing again about the antioxidant….

METHODS: Please reported the initial reference you follow their methodology and not cite your previews article. Also please write a brief presentation of the methodologies you follow.

RESULTS and DISCUSSION. In my opinion, the presented results are not discussed accordingly. The discussion is poorly written and should be revised and re-written well and compare your results with other papers.

CONCLUSION: Well, described

To conclude, reported data are sufficiently presented and commented and the results support sufficiently the author’s conclusion. Therefore, I think that this paper is suitable for publication after the correction of the above observations.

Author Response

Authors are grateful for the contribution of the Reviewer.

ABSTRACT: Please reduce the number of words up to 200 according to the journal’s instructions.

Please omit the “.” After “7th”

Answer:

All comments are made in manuscript.

INTRODUCTION:

Line 65: Please use the same unit methodology and write “85 Kcal / 100 g”

Line 70: “other health-promoting properties” and writing again about the antioxidant….

Answer:

All comments are made in manuscript.

METHODS: Please reported the initial reference you follow their methodology and not cite your previews article. Also please write a brief presentation of the methodologies you follow.

Answer:

All comments are made in manuscript (lines 460-464; 496-524).

RESULTS and DISCUSSION. In my opinion, the presented results are not discussed accordingly. The discussion is poorly written and should be revised and re-written well and compare your results with other papers.

In the results and discussions, the obtained results were compared with those of other authors and fragments of the text were added (lines 145-147; 151-154; 186-189; 197-198; 216-218; 242-245; 250-256; 277-281).

Round 2

Reviewer 1 Report

The authors improved the manuscript. Only minor English corrections are needed before acceptance.

Reviewer 2 Report

The revised manuscript has been improved and covers my observations and I believe that it can be published.